# Indane Based Molecular Motors: UV-Switching Increases Number of Isomers

**DOI:** 10.3390/molecules27196716

**Published:** 2022-10-09

**Authors:** Valeriy P. Shendrikov, Anna S. Alekseeva, Erik F. Kot, Konstantin S. Mineev, Daria S. Tretiakova, Abdulilah Ece, Ivan A. Boldyrev

**Affiliations:** 1Shemyakin-Ovchinnikov Institute of Bioorganic, Chemistry of the Russian Academy of Sciences, GSP-7, Ulitsa Miklukho-Maklaya, 16/10, 117997 Moscow, Russia; 2Higher Chemical College, Mendeleev University of Chemical Technology of Russia, Miusskaya Sq. 9, 125047 Moscow, Russia; 3Department of Pharmaceutical Chemistry, Faculty of Pharmacy, Biruni University, Istanbul 34010, Turkey

**Keywords:** molecular motors, UV-switches, indane, rotamers

## Abstract

We describe azophenylindane based molecular motors (aphin-switches) which have two different rotamers of *trans*-configuration and four different rotamers of *cis*-configuration. The behaviors of these motors were investigated both experimentally and computationally. The conversion of aphin-switch does not yield single isomer but a mixture of these. Although the *trans* to *cis* conversion leads to the increase of the system entropy some of the *cis*-rotamers can directly convert to each other while others should convert via *trans*-configuration. The motion of aphin-switches resembles the work of a mixing machine with indane group serving as a base and phenol group serving as a beater. The aphin-switches presented herein may provide a basis for promising applications in advanced biological systems or particularly in cases where on demand disordering of molecular packing has value, such as lipid bilayers.

## 1. Introduction

Optical switches are a type of molecular motors; they change the conformation (or pass through the open-close reaction) under the effect of light. Photoswitches based on reactions of *E/Z* isomerization include azobenzenes [1] and alkenes (such as stiff-stilbene [2] and particularly sterically hindered alkenes [3]) as well as retinal analogs, derivatives of green fluorescent protein and hemithioindigo [4]. Photoswithes based on the ring open/close reaction include donor-acceptor Stenhouse adducts [5], diarylethene [6,7], spiropyran [8,9], and spirooxazine [8].

Today, the photoswitch research is focused on tailoring the molecular structure of a photoswitch to address the application goal. Impressive gains are obtained in developing the chiral optical materials [10,11], extraction of mechanical work [12] and moving the excitation light to the visible range [13].

One of the applications of photoswitches is to control the lipid bilayer. Such switches are promising membrane permeability regulators or antimicrobial agents inducing the membrane lysis. Starting from the 1990s, lipid chemistry has utilized derivatives of azobenzene as optical switches [14,15]. These optical switches have been used to control the lipid phase segregation behavior in membranes [16,17,18], modulate the protein kinase C [19], TRPV1 receptor [20] or model ion channels [21] activity. Practically, in all the known cases, the optical switches based on simple unsubstituted azobenzene are used in lipid membrane. There has been only a single report on ortho-fluoro derivatives of azobenzene applied [22]. This fact encouraged us to test other structures.

The action of membrane photoswitch is associated with the change of its molecular projection area on lateral stress profile inside the bilayer [23]. Thus, to improve efficacy of membrane photoswitch one should design these with an increased projection area in one of the configurations (either *cis* or *trans*). List of perspective structures of membrane photoswitches [23] includes photoswitches with hydrocarbon cycles conjugated to benzene ring (such as indane) of the azobenzene core. Our motivation was to develop the new photoswitches with the increased projection area (for application to lipid membranes) when we have started to work with isomerizable indane derivatives. While addressing the goal of increasing projection area, the introduction of an additional ring to the azobenzene core dramatically changes the symmetry of the photoswitch and corresponding molecular geometry. This leads to the appearance of additional rotational isomers. While azobenzene has only one isomer in cis configuration, indane based photoswitch has four (Figure 1). Hence, conformational analysis of indane based switches is the subject of the current work. Below we describe synthesis, structure, properties and isomerization of indane based photoswitches.

## 2. Materials and Methods

Starting compounds and silica were obtained from Merck KGaA (Darmstadt, Germany). Solvents were obtained from Ekos-1 (Moscow, Russia) and redistilled prior use. Structures of the obtained compounds were confirmed by the conventional heteronuclear NMR approach. For this purpose, a set of 1D and 2D NMR spectra was recorded on a Bruker Avance III 600 MHz NMR spectrometer equipped with a triple resonance cryogenic TXI probe. We recorded 1D ${}^{1}$H and ${}^{13}$C, 2D homonuclear DQF-COSY, 2D heteronuclear ${}^{1}$H-${}^{13}$C-HSQC (multiplicity edited), ${}^{1}$H-${}^{13}$C-HMBC and ${}^{1}$H-${}^{15}$N-HMBC spectra.

6-Nitroindan-1-one (**2**) and 4-nitroindan-1-one (**3**)

A mixture of sulfuric (14 mL) and nitric (2.7 mL) acids were cooled to −10 ${}^{{}^{\circ}}$C on an ice/salt bath. A solution of indanone (1 g, 7.6 mmol) in 1.1 mL of nitromethane was added in 200 mcl portions under constant stirring. The temperature was kept below −7 ${}^{{}^{\circ}}$C. The adding took 30 min. The mixture was kept additional 15 min on the bath. The sediment has appeared, the mixture has thickened and became light green. The mixture was poured onto ice (200 mL) and extracted with CH${}_{2}$Cl${}_{2}$ (2 × 100 mL). The extract was washed with 5% KHCO${}_{3}$ (2 × 15 mL) and brine, dried over Na${}_{2}$SO${}_{4}$ and evaporated. The residue was purified by chromatography on silica 60 (63–200 mm) using petroleum ether : ethyl acetate step gradient (200:40, 180:60, 160:80). The chromatography yields both 4-nitroindane-1-one (15%) and 6-nitroindane-1-one (63%) as beige and white crystals respectively. Overall yield 78%. 4-nitroindane-1-one **3**: $t_{m}$ 99–100 ${}^{{}^{\circ}}$C (lit. 100–101 ${}^{{}^{\circ}}$C) ${}^{1}$H-NMR (CDCl${}_{3}$) δ ppm: 2.88 (m, 2H, CH${}_{2}$), 3.73 (m, 2H, CH${}_{2}$), 7.69 (dd, 1H, arom), 8.16 (d, 1H, arom), 8.55 (d, 1H, arom). 6-nitroindane-1-one **2**: $t_{m}$ 72–73 ${}^{{}^{\circ}}$C (lit. 74 ${}^{{}^{\circ}}$C) ${}^{1}$H-NMR (CDCl${}_{3}$) δ ppm: 2.91 (m, 2H, CH${}_{2}$), 3.35 (m, 2H, CH${}_{2}$), 7.74 (d, *J* = 8.5 Hz, 1H, arom), 8.52 (dd, *J* = 2.2 Hz, *J* = 8.5 Hz, 1H, arom), 8.65 (d, *J* = 2.2 Hz, 1H, arom).

Ethyl-2-[6-nitroindan-1-ylidene]acetate (**4**) and ethyl 2-[4-nitroindan-1-ylidene]acetate (**5**)

An ampule was loaded with 5 mL of degassed toluene, 0.7 g (4 mmol) of nitroketone and 1.67 g (4.8 mmol) of finely ground ethyl (triphenylphosphorany-lidene) acetate, flushed with nitrogen, sealed and heated to 130 ${}^{{}^{\circ}}$C for 16 h. Ampule was cooled down and opened. The content was evaporated, redissolved in CH${}_{2}$Cl${}_{2}$ and purified by chromatography on silica 60 (63–200 mm) using petroleum ether : ethyl acetate step gradient (9:1, 8:2) for 4-nitro compound **5** and (9:1, 8:2, 3:2, 1:1) for 6-nitro compound **4**. Yields are 60–64%. ethyl 2-[4-nitroindan-1-ylidene]acetate **5** (yellow crystals). ${}^{1}$H NMR (CDCl${}_{3}$) δ ppm: 1.42 (m, 6H, CH${}_{3}$), 4.34 (dq, 4H,CH${}_{2}$), 6.16 (s, 1H, =CH,E-isomer), 6.52 (t, 1H,Z-isomer), 3.25 (m, 2H, CH${}_{2}$,E-isomer), 3.48 (m, 2H, CH${}_{2}$,E-isomer), 3.09 (m, 2H, CH${}_{2}$,Z-isomer), 3.16 (m, 2H, CH${}_{2}$,Z-isomer), 7.51 (d, 1H,H-arom), 7.56 (d, 1H, arom), 8.28 (d, 2H, arom), 8.49 (d, 1H, arom), 9.80 (d, 1H, arom). *Z/E* ratio is 4/5. ethyl 2-[6-nitroindan-1-ylidene]acetate **4** (yellow crystals). ${}^{1}$H NMR (CDCl${}_{3}$) δ ppm: 1.41 (m, 6H, CH${}_{3}$), 4.29 (m, 4H, CH${}_{2}$), 6.16 (t, 1H, =CH,E-isomer), 6.47 (t, 1H, =CH, Z-isomer), 3.08 (ddd, 1H, CH${}_{2}$), 3.46 (ddd, 2H,CH${}_{2}$), 3.55 (ddd, 2H,CH${}_{2}$), 3.64 (ddd, 1H, CH${}_{2}$), 3.70 (d, 1H, CH${}_{2}$), 3.99 (d, 1H, CH${}_{2}$), 9.28 (d, 1H, arom), 7.96 (d, 1H, arom), 8.28 (dd, 2H, arom), 7.72 (d, 1H, arom), 8.15 (d, 1H, arom). ${}^{13}$C NMR (CDCl${}_{3}$) δ ppm: 125.4, 116.9, 114.1, 110.8, 60.4, 60.1, 35.6, 31.4, 30.9, 29.8 *Z/E* ratio is 1/1.

4-Aphin and 6-Aphin

Compounds were prepared in two step (reduction and azocoupling) procedure without purification of intermediate amines. Reduction: 125 mg of nitroolefin (**4** or **5**), 11 mg of 5% Pd/C were mixed with 8 mL of ethanol and the reaction flask was connected to a hydrogen-filled balloon. The mixture was stirred for several days. Over the period, both the double bond and the nitro group were reduced. The reaction was traced by TLS until neither starting compound, nor intermediate aminoolefin were detected. The reaction mass was passed through Kieselguhr pack and evaporated. The residue (amine **6** or **7**) was used as is. Azocoupling: 118 mg of amine **6** or **7** and 0.21 g TosOH were dissolved in 3.5 mL of CH${}_{3}$CN and cooled to −10 ${}^{{}^{\circ}}$C on an ice/salt bath. NaNO${}_{2}$ solution (0.1 M, 37.3 mg NaNO${}_{2}$ in 5.4 mL H${}_{2}$O) was added in 400 mcl portions. The temperature was kept below −7 ${}^{{}^{\circ}}$C. The pH was range was 3–5. It was adjusted to 4 using Na${}_{2}$CO${}_{3}$ solution prior addition of phenol. The phenol solution (50.88 mg of phenol, 26 mg of NaOH and 0.57 mL of water) was added dropwize under rapid stirring. pH had increased to 7–8. The reaction mixture was kept in the bath for one hour and become bright yellow. Extraction with ethylacetate and chromatography on silica 60 (63–200 mm) using petroleum ether : ethyl acetate step gradient (9:1, 8:1, 8:2, 3:2) yield product in 51% yield. **4-aphin** ${}^{1}$H NMR (CDCl${}_{3}$) δ 7.88 (d, *J* = 8.3 Hz, 2H), 7.58 (d, *J* = 7.6 Hz, 1H), 7.29 (dt, *J* = 14.6, 7.3 Hz, 2H), 6.96 (d, *J* = 8.3 Hz, 2H), 5.63 (s, 1H), 4.23 (q, *J* = 7.1 Hz, 2H), 3.70 (p, *J* = 7.4 Hz, 1H), 3.42 (ddd, *J* = 16.9, 8.6, 5.0 Hz, 1H), 3.24 (dt, *J* = 16.5, 7.9 Hz, 1H), 2.87 –2.75 (m, 1H), 2.57–2.40 (m, 2H), 1.89 (ddt, *J* = 12.6, 8.7, 7.3 Hz, 1H), 1.35–1.24 (m, 6H), 0.98–0.85 (m, 0H). ${}^{13}C$ NMR (CDCl${}_{3}$) δ 172.87, 158.33, 148.82, 147.77, 147.53, 141.74, 127.23, 125.18, 124.87, 116.47, 115.76, 60.55, 41.33, 40.05, 32.39, 29.70, 29.46, 27.77, 14.27. **6-aphin** ${}^{1}$H NMR (CDCl${}_{3}$) δ 7.89–7.83 (m, 2H), 7.74 (dd, *J* = 8.0, 1.8 Hz, 1H), 7.73–7.70 (m, 1H), 7.35 (d, *J* = 7.9 Hz, 1H), 6.98–6.93 (m, 2H), 5.82 (s, 1H), 4.24 (qd, *J* = 7.1, 2.1 Hz, 2H), 3.72–3.64 (m, 1H), 3.02 (ddd, *J* = 16.4, 8.5, 4.7 Hz, 1H), 2.98–2.86 (m, 2H), 2.56–2.44 (m, 2H), 1.90–1.81 (m, 1H), 1.32 (t, *J* = 7.1 Hz, 4H). ${}^{13}C$ NMR (CDCl${}_{3}$) δ 158.28, 152.08, 147.13, 146.86, 146.80, 124.91, 124.79, 123.01, 116.44, 115.79, 60.63, 41.29, 39.90, 32.64, 31.19, 14.27.

### 2.1. Absorption Spectra and UV-Isomerization

Electron absorption spectra of **4-aphin** and **6-aphin** solutions were recorded on an SF 2000 spectrophotometer (“OKB Spectr”, St. Petesrburg, Russia). 1 cm quartz cuvettes were used. The volume of the solution was 2 mL, optical density < 0.8. The samples were irradiated with a Philips PL-S lamp 9W power, emission maximum 365 nm (Actinic BL PL-S 9W/10/2P 1CT/6X10BOX Full product code: 871150095194680). The solutions were intensively mixed with a magnetic stirrer. The solvents were pre-distilled and degassed by nitrogen bubbling through for 10 minutes. The photoisomerization kinetics was calculated under the assumption that the photoisomerization rate does not depend on the concentration of the initial substance, which is true for low concentrations. For data fitting, a zero-order in concentration kinetic equation was used. The intensity of the incident radiation was maintained constant by carrying out kinetic measurements in the same box with a light reflector and a fixed distance between the lamp and the cuvette.

### 2.2. Quantum Chemical Calculations

Geometry optimization was performed as follows: three-dimensional structures generated in ChemSketch were preliminarily optimized at BP86 level with def2-SVP basis set, using COSMO solvent model for EtOH solvent including D4 dispersion correction. The resulting structures were further optimized with the B3LYP/def2-TZVP using the following keywords as solvent model and correction as well: COSMO EtOH, D4, VERYTIGHTOPT. Photospectra were obtained from TD-DFT calculations performed at the B3LYP/def2-TZVPD level of theory, with RIJ-COSX approximation, EtOH solvent, dispersion correction D4. The B3LYP functional with def2-TZVP basis set is used because it is very standard and frequently found in literature, thus allowing us to compare data from different sources. The Gibbs energy of the structures was calculated as follows. The calculation of vibrational frequencies, translational *S*(trans), rotational *S*(rot) and vibrational *S*(vib) components of entropy *S*, thermal translational correction, thermal rotational correction and thermal vibrational correction components of internal energy and correction for fluctuations at zero temperature (ZPE) were carried out using the B3LYP/def2-TZVP level of theory, D4 dispersion correction, COSMO EtOH solvent model, and RIJCOSX approximation. These yielded $G_{RIJCOSX}$—Gibbs energy without electronic contribution. The electronic energy $E_{el}$ was taken from the similar calculation, but without using an RIJCOSX approximation. The values of $G_{RIJCOSX}$ and $E_{el}$ were summarized to obtain the final value of Gibbs energy *G* in Hartree. The populations of different rotamers were calculated using the Boltzmann equation and Gibbs energies. Energy barriers to rotation were calculated from the computations at BP86/def2-SVP in EtOH by using the COSMO model D3 as correction. All calculations were carried out using ORCA 4.2.1 program [24].

## 3. Results

Isomers and conformers of molecules investigated in the manuscript are presented in the Figure 1 and optimized geometries of these are decribed below. Reference compound is 4-hydroxyazobenzene (4-hab) (Figure 1 left). Colored inset in Figure 1 represents the 3D structure of *cis*-isomer of the 4-hab. Mirroring of the latter through planes *ox* and *oy* gives the same structure. In contrast to 4-hab, indane based switches 4-aphin (Figure 1 center) and 6-aphin (Figure 1 right) have two conformers in *trans* configuration and 4 conformers in *cis* configuration. This is the consequence of symmetry loss upon introduction of an indane group into the molecular structure.

We name conformers and isomers of aphin switches according to the following: in addition to *cis* and *trans* (which relate to the configuration of the N=N double bond) designations “rot 0”, “rot 180”, “syn” and “anti” are used. Designations “rot 0” and “rot 180” refer to the position of phenol ring close to (rot 0) or far from (rot 180) cyclopentane ring of the indane. Designations “syn” and “anti” refer to the position of the phenol ring on one side of the indane plane with CH${}_{2}$COOEt group (syn) or on opposite sides (anti).

4-aphin and 6-aphin are designed to have two connection points: one is an ester protected carboxy group connected to the cyclopentane ring of the indane core; the other is a phenol hydroxy group. These connection points are introduced for future applications.

The most important feature of the aphin switches is the indane group. The group is introduced to increase the projection area of the switches. The indane group is not entirely planer. Cyclopentane’s CH${}_{2}$ group (at C2 atom) projects out of the plane. The angle at which the CH${}_{2}$ group deviates from the indane plane (according to our calculations) is 26.7 degrees, which is in a good agreement with previously published data for cyclopentene [25]). Two configurations are then possible: one with cyclopentane CH${}_{2}$ staying on the opposite sides of the indane plane with CH${}_{2}$COOEt group, or on the same side. The first one is energetically favorable. It is used throughout the study.

### 3.1. Synthesis

Target compounds were synthesized through a four step procedure (Figure 2) starting from indanone 1. Nitration according to [26]) yielded two isomers: 6-nitro-indanone **2** and 4-nitro-indanone **3** which were separated by column chromatography. Isomer ratio was found to be 4:1 (6-nitro/4-nitro). Each of the nitro-indanones was then subjected to Wittig olefination. This step was performed using toluene as a solvent either under refluxing conditions or heated in sealed ampoule under inert atmosphere. Yield of esters **4** and **5** was around 42–47% regardless of the setup. We assign the yield drop to the substrate enolization, and corresponding gumming [27]. Olefin Z:E ratio was 4:5 for 4- and 1:1 for 6-isomer. Esters **4** and **5** were reduced in ethanol with hydrogen gas on Pd/C catalyst. Resulting amines **6** and **7** were azocoupled (NaNO${}_{2}$/TsOH ) with phenol in water/MeCN mixture (setup similar to black quencher synthesis described in [28]) to yield target compounds 4-aphin and 6-aphin.

### 3.2. Molecular Geometry

#### 3.2.1. *Trans* Isomers

Geometries of molecules under study were obtained through high level quantum chemical calculations. *Trans* configurations of 4-aphin and 6-aphin (R isomers) are presented in Figure 3. Phenol ring lies almost in plane with indane group. Two rotamers are possible. Namely “trans rot 0” and “trans rot 180”.

The *trans* isomers of 4-/6-aphin are close to each other in energy (Table 1). The 4-aphin “rot 180” conformer is slightly more stable than the corresponding “rot 0” conformer (energy difference is around 1 kJ/mol). In contrast to 4-aphin, the 6-aphin “rot 0” is more stable than the corresponding “rot 180” conformer. To change the conformation from “rot 0” to “rot 180” the phenolazo group should be rotated; the energy barrier for the rotation is ca. 30 kJ/mol (Figure 3B and Appendix A). This is 3 times higher than the energy barrier for the rotation about C-C bond in ethanes ([29] and ref. 1–4 therein).

“Rot 0” and “rot 180” rotamers of *trans* isomers 4-/6-aphin do not differ much in dimensions. The distance between connection points (Alternatively distance between phenolic OH and indane C1 group could be compared. This one does not change upon CH${}_{2}$COO fragment rotation. Both distances are presented in Table 1 and Table 2) (distance between phenolic OH and ester COO groups) is around 13 Å for any rotamer (Table 1 and Table 2). The difference between dipole moments of rotamers is around 1D.

*Trans* isomers of both 4- and 6-aphin are not fully planar. Phenol ring deviates from the indane plane and corresponding dihedrals are not equal to neither $0^{{}^{\circ}}$ nor $180^{{}^{\circ}}$. The deviation is bigger for 4-aphin than for 6-aphin (Table 1 and Table 2). The reason could be attributed to the steric hindrance because 4-aphin has cyclopentane group closer to the phenol ring. The proximity to the indane core clearly affects the orientation of the phenol ring.

Both rotamers “rot 0” and “rot 180” coexist simultaneously in solution. 4-aphin rotamers have slightly different ${}^{1}$H NMR spectra. Figure 4 shows the line broadening of aromatic H in phenol and indane groups in the case of 4-aphin. This is due to the H-atoms in 4-aphin rotamers have almost equal but not identical chemical environment (NMR chemical shifts are very close to each other). The alkyl group in the 4-aphin is located close to the rotatable C-N bond. The presence of the alkyl groups influences chemical shifts of the aromatic protons in the indane and phenol cores differently for “rot 0” and “rot 180” rotamer. Regardless, the difference is small. It is interesting that 4-aphin rotamers ratio is not 1:1. One of the rotamers slightly dominates (Figure 4D). Rotamers ratio is 1.06. The fact corresponds to the results of the calculations. According to Gibbs energies of 4-aphin rotamers (Table 1) calculted fractions of 4-aphin rotamers are 73% and 27%. “rot 180” dominates. At the same time the rotamers ratio available from calcultions (≈2.7) is far from those found in NMR.

In contrast to 4-aphin, 6-aphin NMR spectra has no line broadening. Thus, either rotamers are indistinguishable by NMR or there exists only one rotamer. Rotamers could be indistinguishable by NMR due to the too small structural differences between them. In contrast to 4-aphin, 6-aphin has alkyl groups located farther from the rotatable C-N bond. Thus, the influence of the alkyl group on the chemical shifts of the aromatic protons of 6-aphin should be even smaller than for 4-aphin. It could be regarded as negligible. Alternatively, there could be only one rotamer of 6-aphin. The last idea is supported by the results of calculations. One of the rotamers has 93% fraction (Table 2). We beleive that in the case of 6-aphin indistiguishability of rotamers takes place.

#### 3.2.2. *Cis* Isomers

Optimized geometries of *cis* configurations of 4-aphin and 6-aphin (R isomers) are presented in Figure 5. Each has four rotamers with phenol ring projecting out of the indane plane. Geometrical parameters are given in Table 1 and Table 2.

A remarkable difference can be observed between rotamers of *cis* forms of aphin switches. The distance between connection points for *cis* 4-aphin rotamers is only 5.5 Å for “syn rot 0” whereas that for “anti rot 180” is as big as 12.2 Å (Table 1). The same is observed for *cis* 6-aphin: 5.2 Å vs. 12.7 Å (Table 2). Dipole moments are also notably different.

Rotamers do differ in energy (Table 1 and Table 2). The transition between rotamers can be carried out by the rotation around the N=N-C-C bond at the indane fragment. The energy diagrams for 4-aphin and 6-aphin with dihedral near 180${}^{{}^{\circ}}$ (and 0${}^{{}^{\circ}}$) exhibit cusps which correspond to phenol ring passing through the indane plane (Figure 5). The “anti rot 0” and “syn rot 0” 4-aphin rotamers cannot be transformed directly into each other due to the high potential barrier (52–55 kJ/mol, Figure 5B and Appendix A). However, the sequence of transitions “anti rot 0” -> “anti rot 180” -> “syn rot 180” -> “syn rot 0” is possible.

### 3.3. UV-Spectra

The photospectra of aphin switches and reference compound are presented in Figure 6. 4-aphin (Figure 6B) and 6-aphin (Figure 6C) spectra are similar to those of 4-hab (Figure 6D). There are two absorption maxima: an intense one at 340–355 nm (π–π*) and a weak one at 430 nm (*n*–π*). 4-aphin spectrum demonstrates plateau at long wavelength region due to the overlap of electronic transitions. 6-aphin spectrum exhibits separation of electronic transitions typical to 4-hydroxyazobenzenes [30] (compare spectra at Figure 6B–D). The value of the molar extinction coefficient for the π–π* transition is 13,800 (4-aphin) and 21,500 (6-aphin) (Figure 7A,B). These values are below the extinction coefficient of reference compound 4-hab (30,200).

The value of the molar extinction coefficient for the *n*–π* transition is in the range of 800–1800 for 4-aphin (Figure 7C), 300–1500 for 6-aphin (Figure 7D) at 430 nm.

The solvent type and polarity do not affect the value of the absorption coefficient of the π–π* transition, but do change the position of the $\lambda_{max(\pi -\pi *)}$ (Figure 7E,F). An increase of the polarity of the solvent shifts the absorption maximum to the red region. Interestingly the dependence is not linear and the plot $\lambda_{max(\pi -\pi *)}$ vs. dielectric permittivity has an extremum.

Both the intensity and wavelength of the *n*–π* transition depend on the dielectric permittivity. As for the π–π * transition, for *n*–π* transition plot $\lambda_{max(n-\pi *)}$ vs. dielectric permittivity has an extremum. An increase of the polarity the solvent leads to an increase in the adsorbtion intensity. At the same time, the *n*–π* transition of 4-aphin is more intense than that of 6-aphin.

To explain the difference in extinction coefficients of 4-aphin and 6-aphin, we calculated (TD-DFT) values of the dipole transition moments (μ) and strength of the electronic oscillator (*f*) between the states S0–S2 and S0–S1 for both substances. 4-aphin has μ=0.04145 and f=0.2228 for π–π* transition; and $\mu =1\times 10^{-5}$, $f=8.12\times 10^{-5}$ for *n*–π* transition. 6-aphin has μ=0.06284 and f=0.3265 (π–π*); μ≈0, $f=9.18\times 10^{-6}$ (*n*–π*). The ratio of the forces of electronic oscillators $\frac{f_{6-aphin}}{f_{4-aphin}}=1.47$, which is close to the ratio of the molar extinction coefficients for π–π* band (1.55).

The difference in the transition dipole moments is assumed to be due to the difference in the contributions of the C-H bonds of the cyclopentane fragment to the formation of frontier molecular orbitals (Appendix A).

An increase in the intensity of the *n*–π* transition in the 4-aphin isomer correlates with an increase in the contribution made by the orbitals of the C-H bonds of the cyclopentane ring and the pz orbitals of aromatic rings in N-centered n-orbital. This leads to the orbitals mixing, which removes the symmetry ban on the electronic transition (Appendix A).

The photospectrum of the 4-aphin switch is slightly broadened compared to the spectra of 6-aphin and 4-hab (Figiure Figure 6A). To find out the reason of the broadening we calculated the energies of electronic transitions of each rotamer (Appendix A). The obtained values were then summarized (Fractions of rotamers were estimated according to the Boltzmann equation basing on the energy difference between conformers (Table 1 and Table 2)). Both *trans*-6-aphin rotamers have almost the same electronic transition energies; the absorption maxima differ by only 1 nm. On the contrary 4-aphin rotamers have different frontier orbital energies. Corresponding wavelengths differ by 10 nm. The latter is explained by different dihedral angles –C-C-N=N- of 4-aphin conformers (Table 1 and Table 2). Thus, the broadening of the absorption spectrum of 4-aphin relative to the spectra of 6-aphin and 4-hab is caused by the presence of rotamers and the geometrical differences of these. The data correspond to the signal broadening observed in the ${}^{1}$H-NMR spectra of the 4-aphin (Figure 4).

The photospectra of 4-hab and 6-aphin almost completely coincide in the region of 275–400 nm. At the same time their electronic structures are not identical. Decomposition of the spectral curves into Gaussian components is different (Figure 6C,D).

The photospectra of the *cis* forms of 4-aphin and 6-aphin strongly depend on the solvent used. For nonpolar solvents such as heptane and CCl${}_{4}$, the spectrum is characterized by the presence of a flat maximum in the region of 325–360 nm, which contains two Gaussians. The *n*–π* transition is practically suppressed. In polar solvents, there is one maximum at 295 nm, and the intensity of the *n*–π* transition significantly increases.

### 3.4. Photoisomerization

Aphin switches as well as 4-hab are capable for *trans-cis* isomerization under UV radiation and reverse *cis-trans* izomerization when irradiated with blue light, when heated or with time. The isomerization rate was found to be higher in nonpolar solvents such as heptane and carbon tetrachloride. The rate decreased in chloroform, become very slow in ethanol, and finally was practically suppressed in DMF (Figure 8A,B). These findings also supported the reported evidences of solvent effects on the photoisomerization process [31,32].

The rate of photoisomerization decreased in the series 4-aphin > 6-aphin > 4-OH-azo (Figure 8C). This observation can be explained by the Gibbs energy difference between *cis-* and *trans*-configurations. The difference for 4-aphin is around 40 kJ/mol. Meanwhile, it is 52 kJ/mol for 6-aphin. The values of the rate constants of *trans-cis* isomerization are presented in Figure 8C.

## 4. Conclusions

Finally, based on the overall data, the interconversions of 4-aphin and 6-aphin isomers and rotamers are summarized in Figure 9.

The behaviors of newly synthesized azophenylindane based molecular motors (aphin-switches) were elicited from experimental and high level quantum mechanical calculation findings. *Trans* isomers of aphin switches have two stable rotameric forms: “rot-0” and “rot-180”. Both forms coexist in solution and could convert to each other through a relatively high 30 kJ/mol barrier. The excitation of aphin switches with light follows up with *trans-cis* isomerization. The latter yields a mixture of four different rotamers. Thus, the *trans* to *cis* conversion results in an increase of the system entropy. Energy barrier of rotation in *cis* form is lower than in *trans* form (usually 10–20 kJ/mol) except for 4-aphin “syn rot 0” -> “anti rot 0” conversion. The latter form has an energy barrier of rotation of more than 60 kJ/mol. Direct conversion from “syn rot 0” to “anti rot 180” (or vice versa) is not possible for geometrical reasons and should pass through intermediate state (Figure 9). The motion of aphin-switches resembles the work of a mixing machine with indane group serving as a base and phenol group serving as a beater. In *cis* configuration, the 6-aphin has phenol group able for full round rotation. The rotation pathway of *cis*-4-aphin is broken (Figure 9). Once being in *cis*-configuration aphin switches may pass through all rotamers around. This constantly changes the molecular geometry and parameters related with it, e.g., distance between the connection points, and molecular dipole moment. As a result, the system entropy increases. We believe that lipid derivatives based on aphin-switches discussed herein will substitute known [15,16,17,18] azo-benzene derivatives in lipid bilayer associated research.

## Figures and Tables

**Figure 1 molecules-27-06716-f001:**
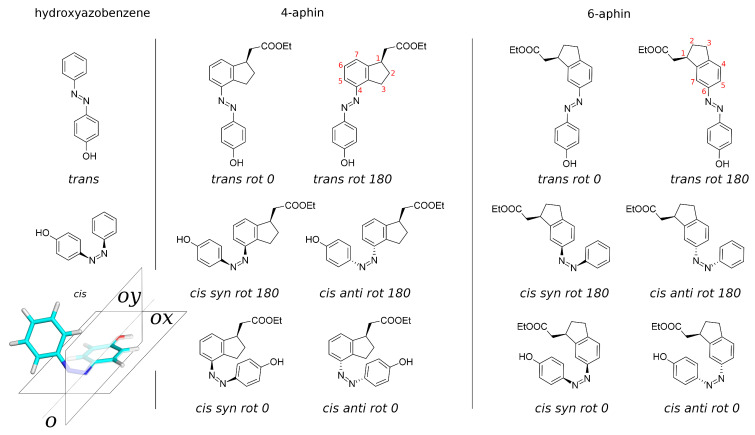
Molecules under study. **Left:** 4-hab (4-hydroxyazobenzene), reference compound. **Center:** 4-aphin (4-(4-hydroxyphenyl) azo indane). **Right:** 6-aphin (6-(4-hydroxyphenyl) azo indane). Stereo-configuration is relative. Red digits represent atom numbering of the indane ring.

**Figure 2 molecules-27-06716-f002:**
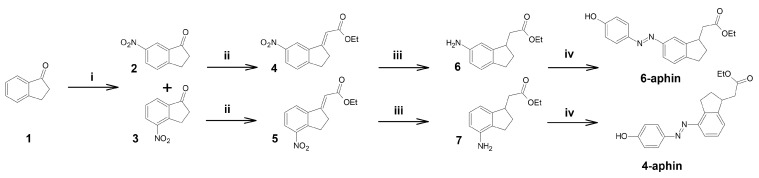
Synthesis of aphin switches. (**i**) HNO${}_{3}$, H${}_{2}$SO${}_{4}$, MeNO${}_{2}$, (**ii**) Ph${}_{3}$PCHCOOEt, Toluene, 140 ${}^{{}^{\circ}}$C, (**iii**) H${}_{2}$, Pd/C, EtOH, (**iv**) NaNO${}_{2}$:TsOH 1:3, phenol+NaOH.

**Figure 3 molecules-27-06716-f003:**
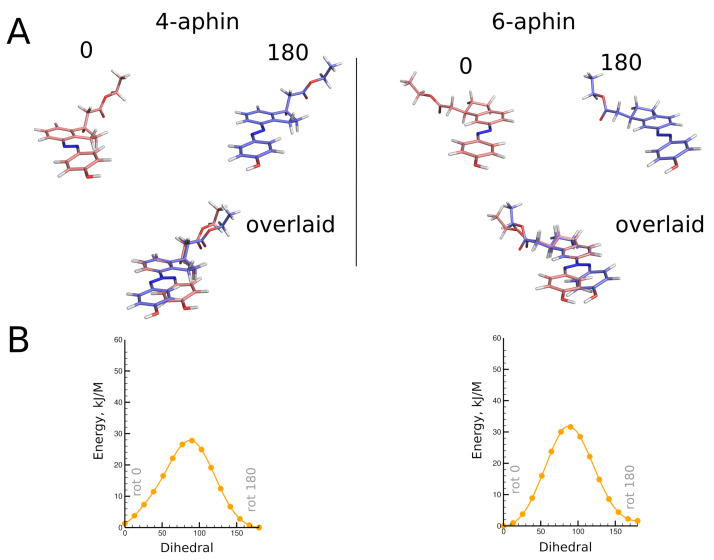
(**A**) **top**: Optimized geometries of *trans* configurations of 4-aphin (**left**) and 6-aphin (**right**) (R isomers). Each has two rotamers. **bottom:** superimposed structures of rotamers. (**B**) Energy barriers for the rotation (C5-C4-N=N dihedral) calculated at BP86 level of theory. (See Appendix A for extremum points calculted at B3LYP level of theory).

**Figure 4 molecules-27-06716-f004:**
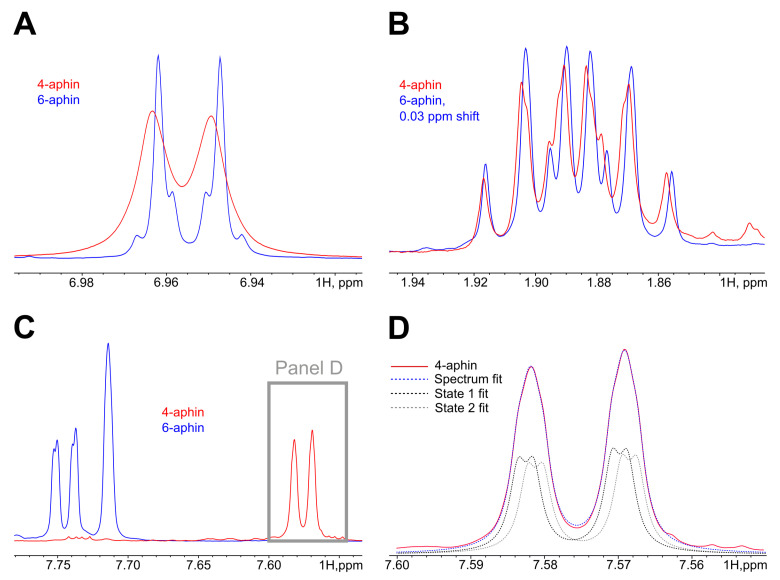
${}^{1}$H-NMR spectra of *trans* isomers 4-aphin (red) and 6-aphin (blue). (**A**) Region 6.9 ppm. Phenol aromatic protons belong to AB system. The AB splitting pattern is observed for 6-aphin but is not observed for 4-aphin. The last signal is broadened. (**B**) Region 1.9 ppm. One of the protons at the C2 of the indane group. Both 4-aphin and 6-aphin have the same splitting pattern; almost no broadening is observed. C2 is far from azo-group. (**C**) Region 7.5–7.8 ppm. Doublets of doublets at 7.74 and 7.58 belong to proton of C5 of the indane group of 6-aphin and 4-aphin correspondingly. It the case of 4-aphin signals from two rotamers are overlapped. (**D**) Overlapped signals shown on panel (**C**) are fitted with two doublet of doublets pattern. The ratio of contributions of doublets (black to gray) is 1.06±0.04.

**Figure 5 molecules-27-06716-f005:**
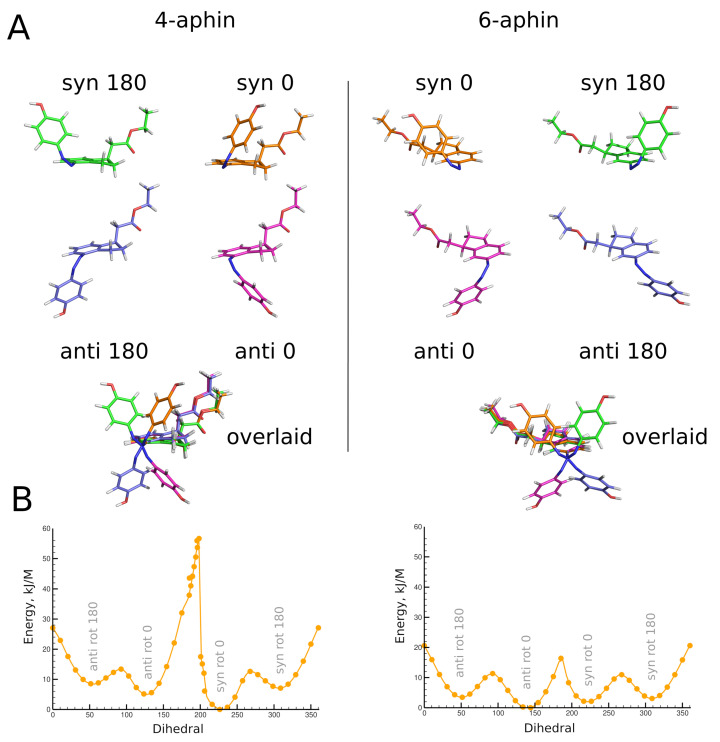
(**A**) **Top:** optimized geometries of *cis* configurations of 4-aphin (**left**) and 6-aphin (**right**) (R isomers). Each has four rotamers. **Bottom:** superimposed structures of rotamers. (**B**) Energy barriers for the rotation (C5-C4-N=N dihedral) calculated at BP86 level of theory. (See Appendix A for extremum points calculted at B3LYP level of theory).

**Figure 6 molecules-27-06716-f006:**
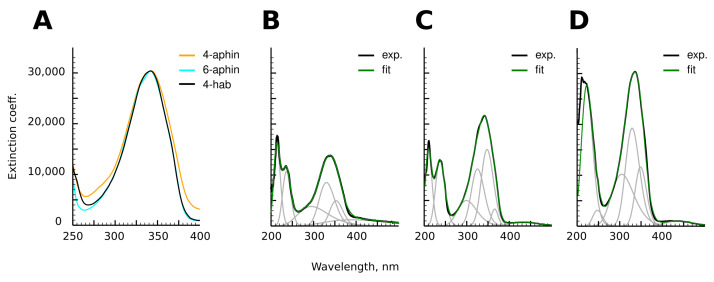
(**A**) Comparison of π–π* transition bands of 4-aphin, 6-aphin and 4-hab. (**B**–**D**). Adsorbance spectra of 4-aphin (**B**), 6-aphin (**C**) and 4-hab (**D**) and their decomposition into Gaussian components.

**Figure 7 molecules-27-06716-f007:**
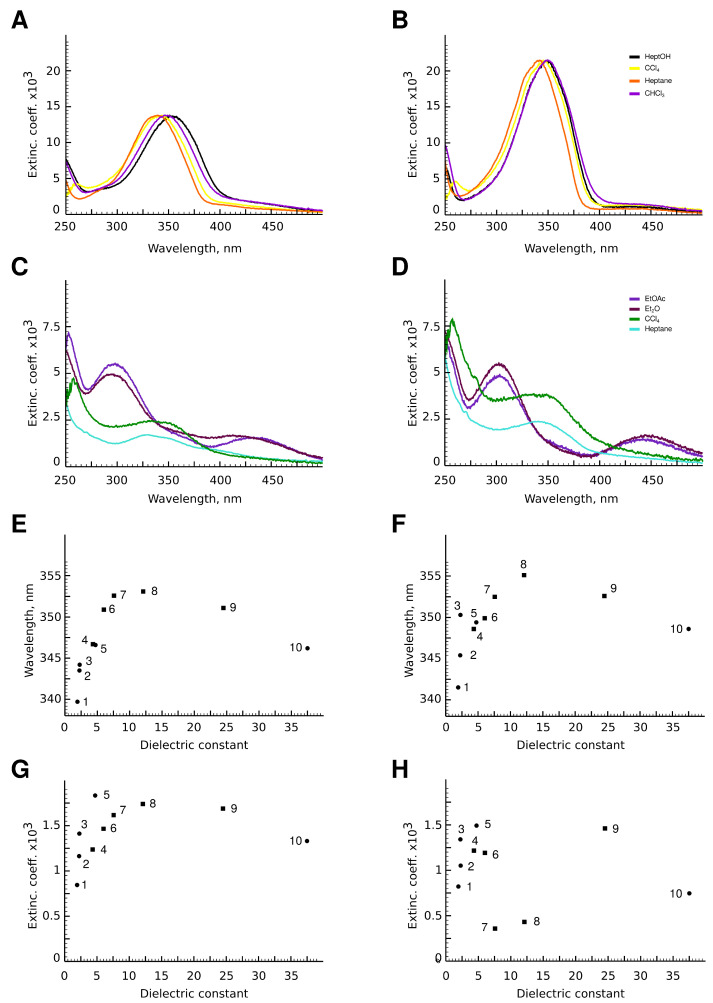
(**A**–**D**) Photospectra of aphin switches. (**A**) 4-aphin *trans* isomer; (**B**) 6-aphin *trans* isomer; (**C**) 4-aphin *cis* isomer; (**D**) 6-aphin *cis* isomer; (**E**,**F**) π–π* absorption maxima $\lambda_{max(pi-pi*)}$ vs. dielectric permeability for 4-aphin (**E**) and 6-aphin (**F**) in Heptane (1), CCl${}_{4}$ (2), benzene (3), Et${}_{2}$O (4), CHCl${}_{3}$ (5), EtOAc (6), THF (7), Heptanol-1 (8), EtOH (9) and MeCN (10). (**G**,**H**) *n*–π* extinction coefficient vs. dielectric permittivity for 4-aphin (**G**) and 6-aphin (**H**). Solvent numbering follows those at panels (**E**,**F**).

**Figure 8 molecules-27-06716-f008:**
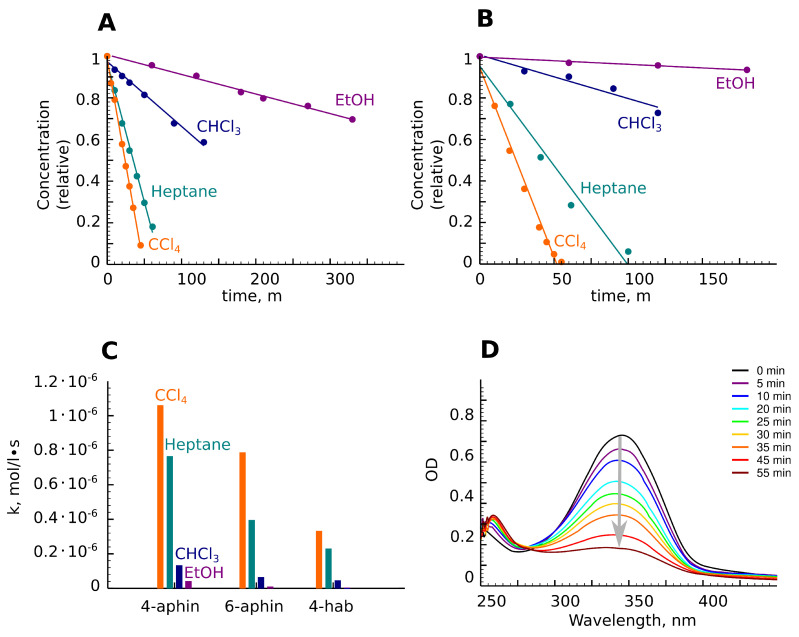
*Trans-cis* photoisomerization. (**A**,**B**) Concentration of *trans*-isomer upon irradiation with time. Experimental points are presented as circles. Fits are presented as lines. (**A**) 4-aphin; (**B**) 6-aphin; (**C**) Constants of isomerization rates. (**D**) UV-spectra of 6-aphin under irradiation in CCl${}_{4}$.

**Figure 9 molecules-27-06716-f009:**
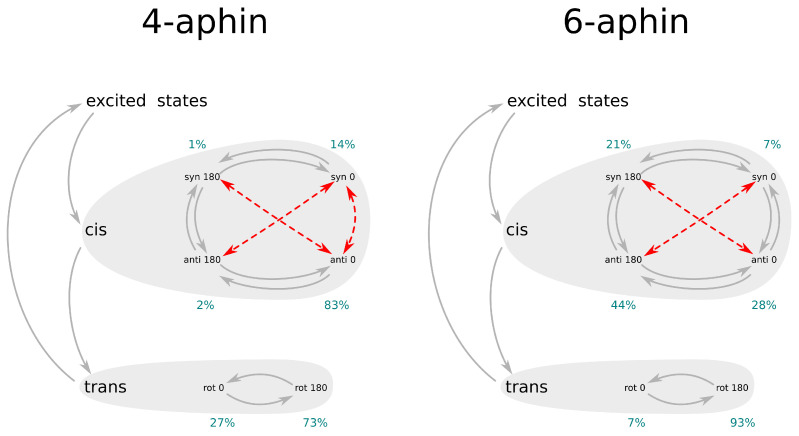
Summary diagram on 4-aphin and 6-aphin isomers and rotamers conversion. Grey arrows represent possible conversions, red arrows represent prohibited conversions. Blue digits represent calculated fractions of rotamers.

**Table 1 molecules-27-06716-t001:** Calculated parameters of the 4-aphin switch (B3LYP) ^1^.

	*trans* rot 180	*trans* rot 0	*cis* anti rot 180	*cis* anti rot 0	*cis* syn rot 180	*cis* syn rot 0
OH–COOEt distance, Å	12.8	12.6	12.2	10.0	9.5	5.5
OH–C1 distance, Å	10.8	10.5	8.6	6.8	8.6	6.5
C5-C4N=N dihedral	5.3${}^{{}^{\circ}}$	−165.8${}^{{}^{\circ}}$	58.6${}^{{}^{\circ}}$	123.0${}^{{}^{\circ}}$	−57.7${}^{{}^{\circ}}$	−128.6${}^{{}^{\circ}}$
Dipole moment, D	3.77	4.34	3.09	3.55	6.48	7.02
*G*, Ha	−1070.80480	−1070.80387	−1070.78583	−1070.78943	−1070.78556	−1070.78775
*S*, Ha	0.06865	0.06839	0.06585	0.07048	0.06586	0.06979
ΔG, kJ/mol	0	2.4	49.7	40.3	50.5	44.7
Fraction, %	73	27	2	83	1	14

^1^ See Figure 1 for atom numbering.

**Table 2 molecules-27-06716-t002:** Calculated parameters of the 6-aphin switch (B3LYP) ^1^.

	*trans* rot 180	*trans* rot 0	*cis* anti rot 180	*cis* anti rot 0	*cis* syn rot 180	*cis* syn rot 0
OH–COOEt distance, Å	13.7	12.7	12.3	9.7	9.7	5.2
OH–C1 distance, Å	11.1	10.7	8.7	6.7	8.7	6.3
C5-C4N=N dihedral	4.6${}^{{}^{\circ}}$	176.4${}^{{}^{\circ}}$	−54.0${}^{{}^{\circ}}$	−134.2${}^{{}^{\circ}}$	53.8${}^{{}^{\circ}}$	135.3${}^{{}^{\circ}}$
Dipole moment, D	2.08	3.19	4.40	4.18	7.13	6.66
*G*, Ha	−1070.80445	−1070.80692	−1070.78706	−1070.78663	−1070.78638	−1070.785292
*S*, Ha	0.06845	0.07097	0.06549	0.06845	0.06839	0.06577503
ΔG, kJ/mol	6.5	0	52.1	53.3	53.9	56.8
Fraction, %	7	93	44	28	21	7

^1^ See Figure 1 for atom numbering.

## Data Availability

Not applicable.

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
