# Peer review of "Indane Based Molecular Motors: UV-Switching Increases Number of Isomers"

_molecules, 2022, doi:10.3390/molecules27196716_

Round 1
Reviewer 1 Report
Comments:
The design of this paper is relatively reasonable and reliable.The azophenylindane based molecular motors (aphin-switches) which have two different rotamers of trans-configuration and four different rotamers of cis-configuration was studied. The retrieval methods of this review is rigorous, and the references are convincing. In this article, authors proposed conversion of aphin-3 switch does not yield single isomer but a mixture of these by both experimentally and computationally study. The trans to cis conversion leads to the increase of the system entropy some of the cis-rotamers can directly convert to each other while others should convert via trans-configuration was proved. Authors think Aphin-switches presented herein may provide a basis for promising applications in advanced biological system. This article has certain theoretical significance and practical value. However, this article still has some problems.
1. In Section 3.4, the rate of photoisomerization decreased in the series 4-aphin>6-aphin>4-OH-azo and authors said it could be explained by the energy gap between cis- and trans- configurations. Please add more details about the energy gap or simulate it by silico.
2. In conclusion, authors said that they believed that aphin-switches discussed in this article will find promising applications in advanced biological systems or particularly in cases where on demand disordering of molecular packing has value, which value did it have in advanced biological systems? How did it play a role? Please state it clearly.
Author Response
Reviewer 1
The design of this paper is relatively reasonable and reliable. The azophenylindane based molecular motors (aphin-switches) which have two different rotamers of trans-configuration and four different rotamers of cis-configuration was studied. The retrieval methods of this review is rigorous, and the references are convincing. In this article, authors proposed conversion of aphin-3 switch does not
yield single isomer but a mixture of these by both experimentally and computationally study. The
trans to cis conversion leads to the increase of the system entropy some of the cis-rotamers can directly
convert to each other while others should convert via trans-configuration was proved. Authors think Aphin-switches presented herein may provide a basis for promising applications in advanced biological
system. This article has certain theoretical significance and practical value. However, this article still has some problems.
Thank you very much for careful reading of the manuscript. To address the comments we have performed additional calculations. The results are incorporated in tables 1 and 2. Figures 3,5 and 9 are modified and discussion section is enriched. Point to point response is below.
1. In Section 3.4, the rate of photoisomerization decreased in the series 4-aphin>6-aphin>4-OH-azo and authors said it could be explained by the energy gap between cis- and trans- configurations. Please add more
details about the energy gap or simulate it by silico.
To address the comment we have performed Gibbs energy calculations. The data are included in tables 1 and 2 and discussed in the body of the ms. The phrase “energy gap” is substituted with exact values of energy difference between cis and trans forms.
2. In conclusion, authors said that they believed that aphin-switches discussed in this article will find promising applications in advanced biological systems or particularly in cases where on demand disordering of molecular packing has value, which value did it have in advanced biological systems? How did it play a role? Please state it clearly.
The romantic statement on “promising application” is removed. Instead we provide citations demonstrating possible applications of aphin switches.

Reviewer 2 Report
Report on the paper entitled "Indane based molecular motors: UV-switching increases
number of isomers", by Valeriy P. Shendrikov, Anna S. Alekseeva, Erick F. Kot, Konstantin S. Mineev, Daria S. Tretiakova, Abdulilah Ece and Ivan A. Boldyrev, submitted to Molecules.
The present paper describes an experimental study of indane based molecular motors. A computational part is added to support the interpretation of the spectra. I will only evaluate this part.
To summarize, the paper is interesting, but suffers from calculations weaknesses which make the paper not acceptable in the present form. Several typos are scattered in the ms.
The calculations are performed with ORCA code and the B3LYP functional with def2-TZVP basis set, COSMO solvation model, Grimme D4 dispersion corrections (ref. should be given), and VERYTIGHTOPT.parameter to fix the accuracy of the calculation.
- The choice of the functional, very standard, should be justified. The review by Mardirossian and Head-Gordon (Molecular Physics, 2017, 115-2315_2372) could be used for the purpose.
- The energy barriers have been calculated with the BP86 functional with def2-TZVP basis set, COSMO solvation model, Grimme D3 dispersion corrections. This is inconsistent with the B3LYP/D4 calculation of the ground state structures: One should, AT LEAST, calculate all the extrema points of Section S4 (min and max) at B3LYP/D4 level.
- The four figures Section S2 should be given in the main text
- The 4-aphin cis does not seem to provide a 360 dihedral degenerate with 0 dihedral. Comment please.
- 4-aphin with dihedral near 180 exhibits a cusp which may be discussed
- 6-aphin with dihedral near 180 exhibits a cusp which may be discussed
- BP86/D3 energies should be added in Tables 1 and 2
The main weakness is the absence of free energies, and entropy contributions of the structures, which are essential when tiny energy differences are involved. This would support (or not) the entropy discussion given in the conclusion.
Minor comments:
- The precision of the calculations (at the level they are performed) cannot deliver accuracy better than 0.5 kJ/mol. Therefore, one digit (at least) has to be removed from the numbers given for the energies.
- Typos lines 23, 231, 299, 301, 301, 311… and other ones.
- pi should be typed π
- Specify that the Table 1 energies are B3LYP calculated.
Author Response
Report on the paper entitled "Indane based molecular motors: UV-switching increases number of isomers", by Valeriy P. Shendrikov, Anna S. Alekseeva, Erick F. Kot, Konstantin S. Mineev, Daria S. Tretiakova, Abdulilah Ece and Ivan A. Boldyrev, submitted to Molecules.
The present paper describes an experimental study of indane based molecular motors. A computational part is added to support the interpretation of the spectra. I will only evaluate this part. To summarize, the paper is interesting, but suffers from calculations weaknesses which make the paper not acceptable in the present form. Several typos are scattered in the ms.
Thank you very much for careful reading of the manuscript. To address the comments we have performed additional calculations. The results are incorporated in tables 1 and 2 and in supplementary material. Figures 3,5 and 9 are modified and discussion section is enriched. Point to point response is below.
The main weakness is the absence of free energies, and entropy contributions of the structures, which are essential when tiny energy differences are involved. This would support (or not) the entropy discussion given in the conclusion.
To address the comment Gibbs energy (including entropy contributions) calculations were performed. The data in presented in tables 1 and 2 and discussed in the body of the manuscript. For the 4-aphin switch we were able to compare calculated (according to Gibbs energy and Boltzmann equation) isomer fractions with those available from the NMR.
The calculations are performed with ORCA code and the B3LYP functional with def2-TZVP basis set, COSMO solvation model, Grimme D4 dispersion corrections (ref. should be given), and VERYTIGHTOPT.parameter to fix the accuracy of the calculation.
- The choice of the functional, very standard, should be justified. The review by Mardirossian and Head-Gordon (Molecular Physics, 2017, 115-2315_2372) could be used for the purpose.
The main reason why we have used B3LYP/D4 is that it is very standard, frequently found in literature and is well documented. We have included the above statement into the Methods section.
Our main goal was to investigate the geometry of molecules and we believe that B3LYP is enough to target the goal. We agree that more sophisticated approach is needed to address energy comparison.
Thank you for pointing us the review by Mardirossian and Head-Gordon. Since we are in the beginning of our calculation team evolution the review will help us a lot.
- The energy barriers have been calculated with the BP86 functional with def2-TZVP basis set, COSMO solvation model, Grimme D3 dispersion corrections. This is inconsistent with the B3LYP/D4 calculation of the ground state structures: One should, AT LEAST, calculate all the extrema points of Section S4 (min and max) at B3LYP/D4 level. The four figures Section S2 should be given in the main text
Figures from section S2 (energy barriers for the rotation) are combined with figures 3 and 5 in the main text of the manuscript. Extremum points are calculated at B3LYP/D4 level. The data is presented in supplementary section S2. Comparison table “BP86 vs B3LYP” is provided for clarity.
- The 4-aphin cis does not seem to provide a 360 dihedral degenerate with 0 dihedral. Comment please.
The figure “energy barrier for the rotation” had an x-axis from 0 to 350 degrees. The point at 360 was absent. The fact appeared to be frustrating. We have modified the figure. Now all point from 0 to 360 are shown.
- 4-aphin with dihedral near 180 exhibits a cusp which may be discussed
- 6-aphin with dihedral near 180 exhibits a cusp which may be discussed
The cusps near 180 (and 0 also) originate from the phenol ring passing through indane plane. The statement is added to the discussion.
- BP86/D3 energies should be added in Tables 1 and 2
Tables 1 and 2 are enriched with Gibbs energy and Entropy. BP86 Energy and B3LYP energy are compared in special table in supplementary section S2.
Minor comments:
- The precision of the calculations (at the level they are performed) cannot deliver accuracy better than 0.5 kJ/mol. Therefore, one digit (at least) has to be removed from the numbers given for the energies.
Too precise digits issue is corrected.
- Typos lines 23, 231, 299, 301, 301, 311… and other ones.
Typos are corrected.
- pi should be typed π
pi is replaced with π
- Specify that the Table 1 energies are B3LYP calculated.
The reference to B3LYP is included in the captions of tables 1 and 2.

Round 2
Reviewer 2 Report
the paper can be published in the present form